# Epigenetic Mechanisms Influencing Epithelial to Mesenchymal Transition in Bladder Cancer

**DOI:** 10.3390/ijms20020297

**Published:** 2019-01-13

**Authors:** Sara Monteiro-Reis, João Lobo, Rui Henrique, Carmen Jerónimo

**Affiliations:** 1Cancer Biology and Epigenetics Group, Research Center, Portuguese Oncology Institute of Porto (CI-IPOP), R. Dr. António Bernardino de Almeida, 4200-072 Porto, Portugal; sara.raquel.reis@ipoporto.min-saude.pt (S.M.-R.); jpedro.lobo@ipoporto.min-saude.pt (J.L.); rmhenrique@icbas.up.pt (R.H.); 2Department of Pathology, Portuguese Oncology Institute of Porto (IPOP), R. Dr. António Bernardino de Almeida, 4200-072 Porto, Portugal; 3Department of Pathology and Molecular Immunology, Institute of Biomedical Sciences Abel Salazar, University of Porto (ICBAS-UP), Rua Jorge Viterbo Ferreira 228, 4050-513 Porto, Portugal

**Keywords:** bladder cancer, EMT, miRNA, lncRNA, epigenetics

## Abstract

Bladder cancer is one of the most incident neoplasms worldwide, and its treatment remains a significant challenge, since the mechanisms underlying disease progression are still poorly understood. The epithelial to mesenchymal transition (EMT) has been proven to play an important role in the tumorigenic process, particularly in cancer cell invasiveness and metastatic potential. Several studies have reported the importance of epigenetic mechanisms and enzymes, which orchestrate them in several features of cancer cells and, specifically, in EMT. In this paper, we discuss the epigenetic enzymes, protein-coding and non-coding genes, and mechanisms altered in the EMT process occurring in bladder cancer cells, as well as its implications, which allows for improved understanding of bladder cancer biology and for the development of novel targeted therapies.

## 1. Introduction

### 1.1. Bladder Cancer

Bladder cancer (BlCa) is the seventh most prevalent cancer worldwide and the second most frequent urological malignancy after prostate cancer. Incidence has been rising in most countries, with an estimated 549,393 new cases diagnosed in 2018 and 990,724 new cases expected in 2040. Therefore, this almost doubled the number. Moreover, BlCa constitutes an important cause of cancer-related death with 199,922 deaths estimated in 2018 and 387,232 predicted for 2040 [1,2]. There is a strong male predominance, approximating a 3:1 ratio, and epidemiological trends track closely the prevalence of tobacco smoking [3]. Similar to other urological malignancies, mortality-to-incidence ratios are higher in underdeveloped countries, which probably reflects different environmental exposures and/or inequalities in healthcare accessibility [4]. Importantly, due to its high prevalence, mortality and, particularly, the propensity for multiple recurrences and/or disease progression and consequent additional treatments, BlCa is the most costly neoplastic disease constituting an important financial burden (costs about €4.9 billion in the European Union, alone, in 2012) [5].

BlCa generally refers to a cancer derived from epithelial layer, the urothelium, which is shared with other organs of the urinary tract and it extends from the renal pelvis to the urethra. Hence, although other much rarer tumor formations occur, its major histological subtype is urothelial carcinoma, which will be the focus of this review. Two major forms of BlCa are acknowledged, differing clinically, pathologically, and molecularly. Non-muscle invasive BlCa (NMIBC, corresponding to 75% to 80% of all cases, disclosing papillary architecture, with the propensity to recur and eventually invade the bladder wall over time) and muscle-invasive BlCa (MIBC, 20% to 25% of all cases, mostly derived from urothelial carcinoma in situ, which constitutes an aggressive disease that invades locally and metastasizes systemically) [6,7].

### 1.2. Epigenetics

During many years, scientists believed that living organisms’ identity was defined by the genetic component of its cells, but, rapidly, it became clear that this could not explain how cells with the same genomic composition could disclose different phenotypes depending on different conditions. Now it is known that the identity of a cell is defined by both genetic and epigenetic patterns with the latter being crucial for fetal development in mammals, as well as cell and tissue differentiation [8,9,10,11]. Epigenetics is defined as the study of heritable modifications of DNA or associated proteins, which carry information related to gene expression during cell division, and currently encompasses all potentially reversible mechanisms that lead to changes in expression regulation without affecting the DNA sequence [12]. The most well-known epigenetic mechanisms comprise four major groups: DNA methylation, histone post-translational modifications or chromatin remodeling, histone variants, and non-coding RNAs’ regulation [11]. These modifications are tightly regulated by several enzymes, which may act isolated or in chromatin remodeling complexes, and grouped according to function. These epigenetic enzymes include: DNA methyltransferases (DNMTs) and demethylases (TETs), histone methyltransferases (HTMs) and demethylases (HDMs), histone acetyltransferases (HATs) and deacetylases (HDACs), and histone ubiquitin ligases (UbLs) and deubiquitinases (dUbs) [12]. 

Cancer cells exhibit a distinct epigenetic landscape and they take advantage of all of the previously mentioned mechanisms to acquire the characteristic malignant features, from transformation to progression [13]. BlCa is no exception. Several studies have associated epigenetic machinery deregulation and this cancer type. Moreover, the potential of epigenetic biomarkers to assist in clinical management of BlCa patients, not only for detection, but also for follow-up, treatment monitoring and prediction of recurrence/progression has been intensively investigated [14,15]. In parallel, efforts have been made to understand how epigenetic mechanisms are involved in the various steps of urothelial carcinogenesis [16,17]. One question remains mostly unanswered. What mechanisms distinguish neoplastic cells with the ability to invade the muscle layer of the bladder, and eventually metastasize, from those that do not have this ability? In fact, epigenetics may help answer this question.

### 1.3. Epithelial to Mesenchymal Transition

The epithelial to mesenchymal transition (EMT) is a multistep process in which epithelial cells acquire a range of mesenchymal characteristics, which enables cell motility and invasiveness [18]. Importantly, these mesenchymal characteristics are reversible, with cells resuming their epithelial phenotype, through a process named mesenchymal to epithelial transition (MET). Recently, the classic concept of EMT, which strictly pointed out to mutually exclusive epithelial or mesenchymal phenotypes, has been challenged by “partial EMT” in which cells may transiently display both epithelial and mesenchymal features, corresponding to an intermediate state of EMT [19,20]. The concept of a partial EMT may be explained by implicating epigenetic regulation of EMT/MET reversibility and cell plasticity. Various factors and cellular environmental conditions are known to induce EMT, by triggering a cascade of signalling pathways that lead to post-transcriptional modification of the most well-known EMT transcription factors (EMT-TFs): Snail, Slug, ZEB1, ZEB2, and TWIST [21,22]. The interplay between the EMT-TFs and various key regulatory proteins and epigenetic enzymes that regulate EMT-TFs themselves, results in overexpression or repression of well-described EMT effectors, such as the cadherin family (CDH1, CDH2, and CDH3) and vimentin [23,24,25,26].

### 1.4. Influence of EMT Major Players in Bladder Cancer

EMT is essential for various physiological processes, including early embryogenesis as well as in cancer. Accordingly, *in vitro* and *in vivo* studies implicated EMT in cell invasion and metastatic potential in several cancer types [19]. Intense research efforts uncovered the major EMT players in epithelial cancers, including BlCa. We performed an *in silico* analysis of The Cancer Genome Atlas (TCGA) database for BlCa (using the online resource cBioPortal for Cancer Genomics [27]), with a user-defined entry set of major EMT players (CDH1, CDH2, CDH3, CTNNB1, GSK3B, MMP2, MUC1, SNAI2, SNAI1, TWIST1, VIM, ZEB1, and ZEB2), and we found that these genes are deregulated in 272/413 (66%) tumors being significantly associated with reduced overall survival (*p* = 0.0098) and disease/progression-free survival (*p* = 0.0279) (Figure 1A,B). Furthermore, the expression levels of mesenchymal markers, like MMP2, VIM, TWIST1, ZEB1, and ZEB2, were significantly higher in stages III/IV when compared to stages I/II (*p* < 0.0001) (Figure 1C–F).

## 2. Epigenetic Enzymes and Mechanisms Altering EMT in Bladder Cancer

### 2.1. Protein-Coding Genes

DNA methylation and chromatin remodelling deregulation in cancer result from aberrant epigenetic enzymes’ activity that ultimately lead to abnormal gene expression, which empowers tumors to quickly evolve. It facilitates invasion and metastasis. Overall, while the importance of these epigenetic enzymes in promoting bladder cancer transformation has been already acknowledged, only a limited number of studies have characterized its role in the context of EMT process in this tumor model. 

One of the epigenetic enzymes involved in EMT is the enhancer of zeste homolog 2 (EZH2), which is a core subunit of the polycomb repressive complex 2 (PRC2) that acts as a chromatin modifier by adding two or three methyl groups at H3K27 residues [28]. In several cancer models, EZH2 was proven to be associated with CDH1 transcriptional silencing and the mesenchymal phenotype [29,30,31]. Using chromatin immunoprecipitation (ChIP), Luo M. et al. demonstrated EZH2 and H3K23me3 enrichment within CDH1 promoter in BlCa cells even though no clues were yet provided on how PRC2 is specifically recruited to CDH1 [32]. Nonetheless, Kottakis et al. suggested that EZH2 might be regulated by FGF-2 upregulation in BlCa cells, which, in turn, upregulates the lysine demethylase 2B (KDM2B) and triggers EZH2 recruitment. The upregulation of these two enzymes is associated with miR-101 transcription repression, due to H3K36 demethylation by KDM2B, and H3K27 trimethylation by EZH2. As a result, and because EZH2 is also post-transcriptionally regulated by miR-101, these events ultimately contribute to EZH2 overexpression in a loop [33,34,35]. Moreover, several EMT-TFs were also found to be overexpressed in these cells, which further supports EZH2 implication in EMT [33,36]. The E2F1 transcription factor and the epigenetic reader BRD4 were also suggested as possible EZH2 regulators in BlCa, but its direct link with EMT and respective TFs is still elusive [37,38]. Importantly, because EZH2 overexpression is common to several tumors, inhibitors for this histone methyltransferase are under evaluation as potential anticancer drugs in phase one and two clinical trials [39]. Nevertheless, just one of the undergoing studies targets BlCa patients, and only those that have unresectable or metastatic disease [40]. The development of new therapies for BlCa is still an unmet need since these tumors have limited treatment options. Specifically, EZH2 inhibition might restrain the progression of non-muscle to muscle invasive disease.

DNA methylation—a covalent modification of DNA, in which a methyl group is transferred from S-adenosylmethionine (SAM) to the fifth carbon of a cytosine‒constitutes a stable and heritable mark frequently associated with the maintenance of a closed chromatin structure, which results in the silencing of repeat elements in the genome and genes’ transcriptional repression [41]. Across the genome, clustered regions of CpG dinucleotides, also known as CpG islands, are often found in genes’ promoter regions. Several cancer-related genes were reported to be regulated by promoter methylation, some of which were implicated in BlCa EMT (Table 1). Among these, serine protease PRSS8 was found to be downregulated by promoter methylation in high-grade BlCa tissues, and its overexpression in cell lines was associated with E-Cadherin upregulation, which suggests an interplay between these two proteins during epithelial differentiation [42,43]. 

Similarly, the Elf5 transcription factor, which is also regulated by methylation in several cellular developmental processes, was associated with EMT in primary BlCa and *in vitro* studies [44,45,46]. Elf5 reduced expression, both at mRNA and protein levels, is associated with disease progression, and, in BlCa cell lines, its downregulation is associated with increased mesenchymal markers, such as Snail, ZEB1, and vimentin. Furthermore, ELF5-silenced BlCa cells exhibited an invasive phenotype, and exposure to the demethylating agent 5-AZA restored ELF5 expression in the same cells, which attenuated its invasion capacity [46]. 

Furthermore, hypermethylation of the growth differentiation factor-15 (GDF15), which is a member of the TGF-β superfamily reported as an urothelial cancer biomarker [51,52], was found to be lower in BlCa cell lines derived from MIBC tumors. Moreover, GDF15-knockdown cells displayed E-Cadherin downregulation while several EMT-TFs were upregulated [48]. Thus, the discovery of epigenetically downregulated genes in MIBC provides new insights about BlCa progression and metastasis.

KLF4, which is a zinc finger transcription factor, is commonly downregulated in several cancers [53,54,55,56] including BlCa [49,50]. Specifically, KLF4 was found to be repressed by promoter methylation [49,50]. Furthermore, (CRISPR)-ON upregulation reduced BlCa cells’ migration, invasion and EMT abilities, which is paralleled by the growth inhibition of tumor xenografts and lung metastasis formation in mice. However, epigenetic editing (e.g., residue specific methylation or demethylation) would be more suitable for assessing the specific role of KLF4 promoter methylation in gene expression regulation [57,58]. The new epigenetic tools available would allow for the clarification of promoter methylation’s regulation of all the previously mentioned genes implicated in BlCa EMT and metastasis.

Several epigenetic mechanisms act synergistically to maintain the epigenetic landscape through a regulation loop in which they simultaneously control protein-coding genes’ expression and other epigenetic players at different regulation levels. Specifically, for BlCa, the oncogene maelstrom (MAEL), frequently upregulated in this malignancy, downregulates the metastasis suppressor MTSS1 gene by recruiting DNMT3B and HDAC1/2 to its promoter. Moreover, MAEL is also targeted by miR-186 and, possibly, by loss of promoter methylation, which constitutes an example of a gene that recruits epigenetic enzymes and is, in turn, regulated by epigenetic mechanisms [47].

### 2.2. Non-Coding RNAs

Non-coding RNAs (ncRNAs) are also involved in the dynamic regulation of EMT-related genes’ expression. There are several ncRNA categories, commonly classified according to their size, including the long ncRNAs (lncRNAs) with more than 200 nt and the small ncRNAs (sncRNAs), which present less than 200 nt [59,60]. ncRNAs, not only directly hinder messenger RNA (mRNA), but also interact (directly or indirectly) with DNMTs, various histone modifying enzymes, and remodelling complexes, which establishes important links between all epigenetic players that modulate gene expression. Therefore, ncRNAs have been implicated in a broad range of biological processes, including proliferation, adhesion, invasion, migration, metastasis, stemness, apoptosis, genomic instability, and, also, EMT, by mediating cell-cell communication (via ncRNA-containing extracellular vesicles), which binds to transcription factors and proteins, DNA methylation regulation, splicing, and scaffolding [61,62].

Among ncRNAs, sncRNAs have been considered the most biologically relevant in the context of EMT. They are involved in post-transcriptional regulation of target RNAs (by forming complexes with proteins of the Argonaute family) with microRNAs being the most intensively studied within this class. Their mature forms are single-stranded and have 20–25 nt in length, which constitutes the final product of a processing pathway involving DROSHA, DICER, and RISC [63]. In fact, *in silico* analysis has shown several up-regulated and downregulated microRNAs that target the most important EMT players associated with aggressive disease [64]. 

Our literature review disclosed 31 different microRNAs, which participate in BlCa EMT regulation [(Table 2), [65,66,67,68,69,70,71,72,73,74,75,76,77,78,79,80,81,82,83,84,85,86,87,88,89,90,91,92]]. Most studies were performed in patients’ samples (*n* = 25) and/or in cell lines (*n* = 31), but some have also tested animal models (*n* = 9). For most microRNAs, the net effect was to counteract an EMT phenomenon (*n* = 25), while only miR92 family/miR92b, miR135a, miR221, miR224, and miR301b were reported to promote EMT. In addition, to a putative value as diagnostic markers, 13 microRNAs were shown to have prognostic and/or predictive value as well, associated with clinicopathological variables such as tumor grade, stage, occurrence of metastases, and patients’ survival.

Some of the most well-studied microRNAs belong to the miR200 family. Their expression has been found to hamper EMT in different tumor models such as breast, prostate, ovarian, and endometrial carcinomas, in part by affecting different EMT players like ZEB1, ZEB2, and E-Cadherin [109,110,111,112,113]. Martínez-Fernández et al. [74] showed that PRC members EZH2 and BMI1 repress miR200 family, which results in EMT activation and aggressive disease, which is in accordance with the association of EZH2 overexpression with high risk for recurrence in NMIBC [114]. These findings support the dynamic regulation and cooperation between protein coding and non-coding RNAs in EMT. Since EZH2 pharmacological inhibition is already available and efficiently increases miR200 in BlCa cell lines, this might constitute a therapeutic opportunity for hindering cancer progression. It has also been reported that epidermal growth factor receptor (EGFR) inhibition may lead to therapeutic resistance due to mesenchymal features. Additionally, miR200c induction (which targets the ERBB receptor feedback inhibitor 1‒ERRFI-1) is effective in restoring sensitivity to EGFR inhibitors, which constitutes another example of pharmacological modulation of EMT that could be translated into clinical practice [65]. Lastly, another member of the miR200 family, miR200b, was demonstrated to target matrix metalloproteinase-16 (MMP16) in BlCa cell lines, which is downregulated by transforming growth factor beta 1 (TGF-β1), previously associated with metastatic potential acquisition. This leads to miR downregulation having a net effect of promoting EMT [75]. In fact, TGF-β1 also cooperates with several other miRs, including miR221. Liu et al. showed that, by targeting STMN1, miR221 facilitates TGF-β1-induced EMT, and that its inhibition resulted in increased levels of epithelial marker E-cadherin and reduction of mesenchymal markers such as vimentin, fibroactin, and N-cadherin [76]. 

A connection between microRNAs and methylation was also reported, which disclosed a feedback loop between DNMT1 and miR-148a-3p [79]. miR-148a-3p, a BlCa tumor suppressor, and an EMT inhibitor, by targeting the ERBB3/AKT2/c-MYC axis, was shown to be downregulated by DNMT1-induced methylation. Moreover, re-expression was observed after treatment with 5-Aza-2’-deoxycytidine (5AZA) [79]. Wu et al. obtained similar findings for miR424 in BlCa cell lines, in vivo models, and patient-derived specimens. DNMT1 inhibition resulted in substantial miR424 upregulation, which, in turn, promoted epithelial characteristics of BlCa cells (changing the relative expression levels of E-cadherin and Twist) and resulted in reduced invasion ability. Additionally, the same authors identified the EGFR-PI3K-AKT axis as the target of miR424, explaining its effect on EMT [77]. Lastly, miR-323a-3p was also implicated in EMT of BlCa cells by targeting MET and SMAD3, which interfered with their regulation of Snail and resulted in the net effect of repressing EMT. On the other hand, miR-323a-3p is downregulated by aberrant methylation of the intergenic differential methylated region (IG-DMR) [87]. In addition, miRs might also be regulated by methylation and this feature might be used for urothelial carcinoma detection in bodily fluids, such as urine [115].

EMT-related miRs have also been demonstrated to impact the resistance to cytotoxic drugs. Furthermore, miR-92 was found to promote EMT (changing the relative expression levels of two of its major players, E-cadherin, and vimentin) by activating glycogen synthase kinase 3 beta (GSK3B) and the Wnt signalling pathway, inducing resistance to cisplatin (increasing BlCa cells viability and decreasing apoptosis upon treatment with cisplatin) [79]. 

Most of the human genome is transcribed into structural ncRNAs. LncRNAs, which include both linear and circular forms (the latter being referred to as circRNAs), display different regulatory functions, according to their cellular location. Whereas nuclear lncRNAs can either sequester transcription factors and recruit chromatin-remodelling complexes to a cell-site (hence impeding transcription), or trigger chromatin-modifying complexes (thus, activating transcription), cytoplasmic lncRNAs modulate RNAs stability and translation, competing with endogenous RNAs (ceRNAs) for microRNA binding. Additionally, having a longer half-life than their linear counterparts, circRNAs may also act as microRNA “sponges” [116,117].

Eleven lncRNAs (ten linear and one circRNA) [97,98,99,100,101,102,103,104,105,106,107,108] have been reported to modulate EMT in BlCa. Contrary to microRNAs, only one lncRNA (TP73-AS1) was implicated in negative regulation of EMT, whereas all the remainder substances promoted its activation. Five of the lncRNAs (including circRNA MYLK and lncRNAs GHET1, Malat1, TP73-AS1, and TUG1) were explored as potential diagnostic and prognostic biomarkers.

CircRNA MYLK was found to function as ceRNA for miR-29a, which, in result, promotes EMT and activates the vascular endothelial growth factor receptor (VEGFR) pathway, which is associated with BlCa progression [97]. Thus, circRNA MYLK modulation might constitute a therapeutic target in combination with anti-VEGF drugs such as bevacizumab. Moreover, Lv et al. [100] have shown that lncRNA H19 also functions as a ceRNA for miR-29b-3p, which is another member of the miR29 family. Therefore, this allows for the expression of target DNMT3B, reprograms DNA methylation patterns, and promotes EMT (through Twist, vimentin, and MMP9 upregulation and E-cadherin downregulation) and metastasis.

Non-coding RNAs may modulate not only the response to systemic treatments, but also to local therapies such as radiotherapy. Tan et al. [104] showed that miR145’s downmodulation by lncRNA TUG1 associated with EMT and radio-resistance due to its action on the ZEB2 axis. Targeting this lncRNA might re-sensitize BlCa to radiotherapy, which results in a better patient response and outcome.

Furthermore, TGF-β1 leads to overexpression of lncRNA malat1, which is associated with suppressor of zeste 12 (suz12), decreases E-cadherin, and increases N-cadherin and fibronectin expression levels [101]. Moreover, another lncRNA-ZEB2NAT—was shown to be essential for the role of TGF-β1-secreting cancer associated fibroblasts (CAFs) in promoting EMT in BlCa cells. Zhuang et al. elegantly showed that CAFs induce EMT by activating the TGF-β1/ZEB2NAT/ZEB2 axis, whereas ZEB2NAT inhibition reduced ZEB2 expression levels and inhibited BlCa cells invasion capacity [108].

Several lncRNAs might target the same microRNA and the same lncRNA may influence more than one microRNA simultaneously. Such is the case of lncRNA UCA1, which induces EMT either by targeting miR145 or miR143 [105,106]. These studies suggest that lncRNAs might be implicated in EMT by interfering with several pathways through various regulatory functions, due to their redundancy.

## 3. Conclusions

As discussed in this review, epigenetic mechanisms and connected enzymes are intrinsically involved in the various steps of EMT in BlCa cells, which acts in concert and controlling EMT-TFs as well as several upstream targets (Figure 2). All the studies published to the date illuminate the way for the development of specific anti-cancer drugs, which could abrogate EMT by targeting epigenetic enzymes and genes regulated by these reversible modifications. 

Nevertheless, the epigenetic regulation of EMT requires further investigation to provide clinically useful information for BlCa patient management.

## Figures and Tables

**Figure 1 ijms-20-00297-f001:**
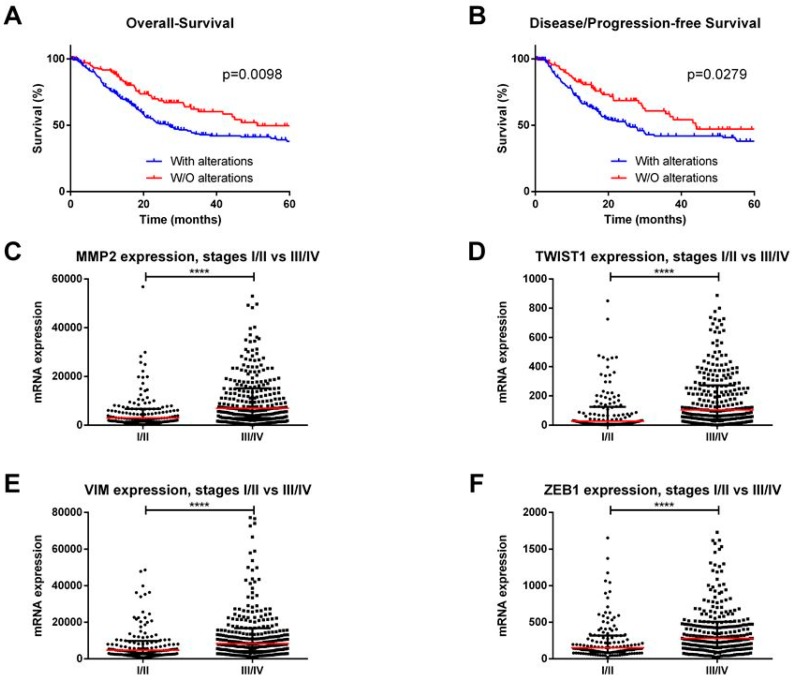
*In silico* analysis of The Cancer Genome Atlas database for bladder cancer (using the online resource cBioPortal for Cancer Genomics). (**A**) Overall and (**B**) Disease/Progression-free survival curves according to major EMT players’ alterations. (**C**) MMP2, (**D**) TWIST1, (**E**) VIM, and (**F**) ZEB1 transcript levels in stages I/II vs. III/IV bladder cancer cases. **** *p* < 0.0001.

**Figure 2 ijms-20-00297-f002:**
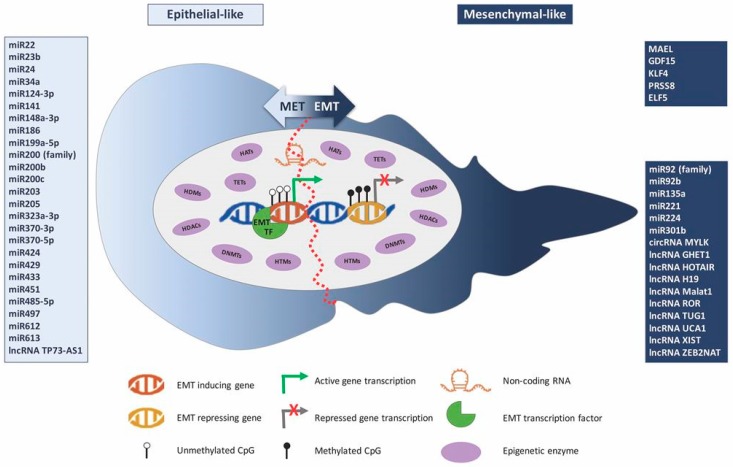
Epigenetic mechanisms’ interplay with the epithelial-to-mesenchymal transition process in bladder cancer.

**Table 1 ijms-20-00297-t001:** Epigenetically modulated protein-coding genes implicated in Bladder Cancer EMT.

Gene	Expression in BlCa	Effect on EMT	Epigenetic Regulation	Sample Type and Size	Author
*MAEL*	Upregulated	↑EMT (↓ECAD, ↓β-catenin, ↑Fibronectin, ↑VIM) Recruitment of DNMT3B and HDAC1/2 to MTSS1 promoter)	Downregulated by miR186	184 primary tumors, *in vitro* and *in vivo* assays	Li, X.D., 2016 [47]
*GDF15*	Downregulated	↓EMT (knockdown cells with ↓ECAD, ↑NCAD, ↑Snail, ↑Slug)	Upregulated by demethylation	*In vivo* assays	Tsui, K.H. and Hsu, S.Y., 2015 [48]
*KLF4*	Downregulated	↓EMT (↑ECAD, ↓NCAD, ↓ β-catenin, ↓VIM, ↓Snail, ↓Slug)	Promoter methylation; Upregulated by 5AZA treatment	139 non-muscle invasive primary tumors, *in vitro* and *in vivo* assays	Li, H. and Wang, J., 2013 [49]
		↓EMT (Upregulation)	Promoter methylation confirmed by BSP	*In vitro* assays	Xu, X., 2017 [50]
*PRSS8*	Downregulated	↓EMT (↑ECAD in cells with forced PRSS8 expression)	Promoter methylation. Upregulated by 5AZA and TSA treatment	40 primary tumors and *in vivo* assays	Chen, L.M., 2009 [43]
*ELF5*	Downregulated	↓EMT (↑ECAD, ↓NCAD, ↓VIM, ↓Snail, ↓ZEB1)	Promoter methylation. Upregulated by 5AZA treatment	182 FFPE + 50 FF primary tumors and *in vivo* assays	Wu, B., 2015 [46]

**Abbreviations:** 5AZA—5-Azacytidine. BlCa—bladder cancer. BSP—Bisulfite sequencing. EMT—epithelial to mesenchymal transition. FF—Fresh-frozen. FFPE—Formalin-fixed paraffin-embedded. miR—microRNA. TSA—Trichostatin A.

**Table 2 ijms-20-00297-t002:** Non-coding RNAs associated with EMT in bladder cancer.

Non-Coding RNA	Effect on EMT (and Others)	Main Regulators	Main Targets/Pathways	Sample Type and Size	Author
***Small Non-Coding RNAs***					
**miR22**	↓EMT, diagnostic value (↓ in tumor, vs. normal)		Snail and MAPK/Slug/VIM	13 primary tumors, *in vitro* and *in vivo* assays	Xu, M., 2018 [90]
**miR23b**	↓EMT, diagnostic (↓ in tumor vs. normal) and prognostic (↑OS) value		ZEB1	20 primary tumors and *in vivo* assays	Majid, S., 2013 [68]
**miR24**	↓EMT, diagnostic value (↓ in tumor, vs. normal)		CARMA3	*In vitro* assays	Zhang, S., 2015 [71]
**miR34a**	↓EMT, diagnostic value (↓ in tumor, vs. normal)		CD44	8 primary tumors, *in vitro* and *in vivo* assays	Yu, G., 2014 [72]
**miR92 (family)**	↑EMT, diagnostic (↑ in tumor vs. normal) value, induces cisplatin resistance		GSK-3β/ Wnt/c-myc/MMP7	20 primary tumors and *In vitro* assays	Wang, H., 2016 [79]
**miR92b**	↑EMT		DAB2IP	*In vitro assays*	Huang, J., 2016 [80]
**miR-124-3p**	↓EMT, diagnostic value (↓ in tumor, vs. normal)		ROCK1, MMP2, MMP9	13 primary tumors and *in vitro* assays	Xu, X., 2013 [66]
**miR135a**	↑EMT		GSK-3β	165 primary tumors and *in vitro* assays	Mao, X.W., 2018 [91]
**miR141**	↓EMT, prognostic value (LN metastases)		MMP2 and 9, Vimentin, N-Cadherin, E-Cadherin	30 primary tumors, 78 urine samples and *in vitro* assays	Liu, W. and Qi, L., 2015 [93]
**miR-148a-3p**	↓EMT, diagnostic value (↓ in tumor, vs. normal)	↓expression mediated by DNA methylation (DNMT1) – ↑expression with 5AZA	ERBB3-AKT2-c-myc/SNAIL axis	59 primary tumors, *in vitro and in vivo* assays	Wang, X., 2016 [82]
**miR186**	↓EMT, diagnostic value (↓ in tumor, vs. normal)		NSBP1	20 primary tumors and *in vitro* assays	Yao, K., 2015 [73]
**miR-199a-5p**	↓EMT, diagnostic (↓ in tumor vs. normal) and prognostic (stage, grade) value		CCR7, MMP9	40 primary tumors and *in vitro* assays	Zhou, M., 2016 [81]
**miR200 (family)**	↓EMT, prognostic value (↑ survival)	↓expression mediated by EZH2 and BMI-1	BMI-1, ZEB1, ZEB2	87 primary tumors and *in vitro* assays	Martínez-Fernández, M. and Duenas, M., 2015 [74]
	↓EMT and proliferation, diagnostic (↓ in tumor vs. normal) and prognostic (↑ survival) value		BMI-1 and E2F3	15 primary tumors and *in vitro* assays	Liu, L., 2014 [69]
**miR200b**	↓EMT, prognostic value (LN metastases)		MMP2 and 9, Vimentin, N-Cadherin, E-Cadherin	30 primary tumors, 78 urine samples and *in vitro* assays	Liu, W. and Qi, L., 2015 [93]
	↓EMT	↓expression mediated by TGF-β1	MMP16	*In vitro* assays	Chen, M.F. and Zeng, F., 2014 [75]
**miR200c**	↓EMT, restores sensitivity to EGFR inhibitors		ZEB1, ZEB2 and ERRFI-1	*In vitro* assays	Adam, L., 2009 [65]
**miR203**	↓EMT, diagnostic value (↓ in tumor, vs. normal)		Twist1	24 primary tumors and *in vitro* assays	Shen, J., 2017 [85]
**miR205**	↓EMT, poor prognosis	↑expression mediated by p63 isoform ΔNp63α	ZEB1, ZEB2	98 primary tumors and *in vitro* assays	Tran, M.N., 2013 [67]
**miR221**	↑EMT	↑expression mediated by TGF-β1	STMN1	*In vitro* assays	Liu, J., 2015 [76]
**miR224**	↑EMT, diagnostic (↑ in tumor vs. normal) and prognostic (stage, metastases, ↓survival) value		SUFU/Hedgehog pathway	97 primary tumors, *in vitro* and *in vivo* assays	Miao, X., Gao, H. and Liu, S., 2018 [86]
**miR301b**	↑EMT, diagnostic value (↑ in tumor, vs. normal)		EGR1	*In vitro* assays	Yan, L., 2017 [94]
**miR-323a-3p**	↓EMT, diagnostic (↓ in tumor vs. normal) and prognostic (↑OS) value	↓expression mediated by methylation of IG-DMR	Met/SMAD3/Snail	9 primary tumors and *in vivo* assays	Li, J., 2017 [87]
**miR-370-3p**	↓EMT		Wnt7a	41 primary tumors *in vitro* and *in vivo* assays	Huang, X. and Zhu, H., 2018 [92]
**miR-370-5p**	↓EMT		p21	*In vitro* assays	Wang, C., 2016 [95]
**miR424**	↓EMT, diagnostic (↓ in tumor vs. normal) and prognostic (stage, ↑OS and DFS) value	↓expression mediated by DNMT1	EGFR pathway	124 primary tumors, *in vitro* and *in vivo* assays	Wu, C.T., 2015 [77]
**miR429**	↓EMT		ZEB1/βcatenin axis	*In vitro* assays	Wu, C.L., 2016 [83]
**miR433**	↓EMT, diagnostic value (↓ in tumor, vs. normal)		c-Met/CREB1-Akt/GSK-3β/Snail	13 primary tumors and *in vitro* assays	Xu, X., 2016 [84]
**miR451**	↓EMT, diagnostic (↓ in tumor vs. normal) and prognostic (grade and stage) value		E-Cadherin, N-Cadherin	40 primary tumors and *in vitro* assays	Zeng, T. and Peng, L., 2014 [70]
**miR-485-5p**	↓EMT, diagnostic value (↓ in tumor vs. normal)		HMGA2	15 primary tumors and *in vitro* assays	Chen, Z., 2015 [78]
**miR497**	↓EMT, diagnostic (↓ in tumor vs. normal) and prognostic (stage, metastases) value		E-Cadherin, Vimentin	50 primary tumors and *in vitro* assays	Wei, Z., 2017 [88]
**miR612**	↓EMT, diagnostic (↓ in tumor vs. normal) and prognostic (stage, metastases) value		ME1	46 primary tumors and *in vitro* assays	Liu, M. and Chen, Y., 2018 [96]
**miR613**	↓EMT, diagnostic value (↓ in tumor vs. normal)		SphK1	35 primary tumors and *in vitro* assays	Yu, H., 2017 [89]
***Long non-coding RNAs***					
**circRNA MYLK**	↑EMT, prognostic value (stage, grade)		miR29a/VEGFA/VEGFR2 axis	32 primary tumors, *in vitro* and *in vivo* assays	Zhong, Z., 2017 [97]
**lncRNA GHET1**	↑EMT, diagnostic (↑ in tumor vs. normal) and prognostic (grade, stage, metastases, ↓OS) value		E-Cadherin, Vimentin, Fibronectin, Slug, Twist, Snail, ZEB1	80 primary tumors and *in vitro* assays	Li, L.J., 2014 [98]
**lncRNA HOTAIR**	↑EMT		Various EMT players	10 primary tumors and *in vitro* assays	Berrondo, C., 2016 [99]
**lncRNA H19**	↑EMT, diagnostic value (↑ in tumor vs. normal)		miR-29b-3p/DNMT3B axis	35 primary tumors, *in vitro* and *in vivo* assays	Lv, M., 2017 [100]
**lncRNA Malat1**	↑EMT, poor prognosis	↑expression mediated by TGF-β	suz12	95 primary tumors, *in vitro* and *in vivo* assays	Fan, Y., 2014 [101]
**lncRNA ROR**	↑EMT, diagnostic value (↑ in tumor vs. normal)		ZEB1	36 primary tumors and *in vitro* assays	Chen, Y., 2017 [102]
**lncRNA TP73-AS1**	↓EMT, diagnostic (↓ in tumor vs. normal) and prognostic (↑OS and PFS) value		Various EMT players	128 primary tumors and *in vitro* assays	Tuo, Z., 2018 [103]
**lncRNA TUG1**	↑EMT, diagnostic (↑ in tumor vs. normal) and prognostic (stage, ↓OS) value, promotes radio-resistance		miR145/ ZEB2 axis	54 primary tumors, *in vitro* and *in vivo* assays	Tan, J., 2015 [104]
**lncRNA UCA1**	↑EMT		miR145-ZEB1/2-FSCN1 axis	*In vitro* assays	Xue, M., 2016 [105]
			miR143/HMGB1	52 primary tumors and *in vitro* assays	Luo, J., 2017 [106]
**lncRNA XIST**	↑EMT		miR200c	*In vitro* and *in vivo* assays	Xu, R., 2018 [107]
**lncRNA ZEB2NAT**	↑EMT, diagnostic value (↑ in tumor vs. normal)	↑expression mediated by TGF-β1	ZEB2	30 primary tumors and *in vitro* assays	Zhuang, J. and Lu, Q., 2015 [108]

**Abbreviations:** DFS—disease-free survival. EMT—epithelial to mesenchymal transition. lncRNA—long non-coding RNA. miR—microRNA. OS—overall survival.

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
