# Peer review of "Epigenetic Mechanisms Influencing Epithelial to Mesenchymal Transition in Bladder Cancer"

_ijms, 2019, doi:10.3390/ijms20020297_

Round 1
Reviewer 1 Report
This review describes epigenetics-related mechanisms of epithelial to mesenchymal transition (EMT) in urinary bladder cancer. Presented article is of great interest, because of its novelty and also very interesting and clear description of the potential epigenetic mechanisms of EMT in urinary bladder cancer ethiology and their clinical relevance. Nevertheless, these state-of-the art findings need minor corrections and clarifications before publication.
As article describes mainly promoter methylation of EMT-associated genes (Table 1) and ncRNA (microRNA and lncRNA) effect on EMT-associated genes (Table 2), I suggest not to underline “the epigenetic enzymes” (see Abstract, 2. Epigenetic enzymes and mechanisms…, page 4 and elsewhere in the text). In fact, there is only 1 paragraph (page 4, from 129th row), where histones modifications lead by selected epigenetic enzymes are described. Thus, I also suggest to change the title of the section.
According to findings presented in Table 1, I suggest to widely describe promoter region methylation and its consequences for gene expression in the section 1.2. Epigenetics (page 2, from 51st row).
Please, provide selection criteria for EMT-associated genes in TCGA database retrieving (1.4. EMT and epigenetics…, page 3). It is good idea to present EMT genes/proteins classification, e.g. markers, transcription factors, signaling pathway-related. I suggest to give the full name of all described EMT players and EMT targets; e.g. in figure and table legend.
Please, rethink the section title: (1.4. EMT and epigenetics in bladder cancer, page 3). Do you really discuss EMT and epigenetics in bladder cancer?
I suggest to remove the selection criteria of EMT and epigenetics related publications (page 3, from 113th row). In my opinion this information seems to be redundant. Unless you have performed critical review. In this way, you should present number of articles you have taken for analysis and articles you have removed.
Table 1. Please, give explanation of: 5AZA, TSA, BSP, FFPE, FF.
Table 1 and Table 2. In author or Ref. column you should provide also appropriate number according to references list.
Author Response
This review describes epigenetics-related mechanisms of epithelial to mesenchymal transition (EMT) in urinary bladder cancer. Presented article is of great interest, because of its novelty and also very interesting and clear description of the potential epigenetic mechanisms of EMT in urinary bladder cancer ethiology and their clinical relevance. Nevertheless, these state-of-the art findings need minor corrections and clarifications before publication.
As article describes mainly promoter methylation of EMT-associated genes (Table 1) and ncRNA (microRNA and lncRNA) effect on EMT-associated genes (Table 2), I suggest not to underline “the epigenetic enzymes” (see Abstract, 2. Epigenetic enzymes and mechanisms…, page 4 and elsewhere in the text). In fact, there is only 1 paragraph (page 4, from 129th row), where histones modifications lead by selected epigenetic enzymes are described. Thus, I also suggest to change the title of the section.
R: Our main goal was to describe the epigenetic enzymes and mechanisms that have been already reported to affect EMT in bladder cancer (e.g. the title of section 2). There is no doubt that histone modifications are one of the major mechanisms of epigenetic modulation of gene expression/repression, however, based on available publications, promoter methylation and non-coding RNAs seem to be the major players. Notwithstanding, all epigenetic mechanisms found in the selected studies, including histone PTMs, were described in the current review.
According to findings presented in Table 1, I suggest to widely describe promoter region methylation and its consequences for gene expression in the section 1.2. Epigenetics (page 2, from 51st row).
R: The information was added as suggested, but included to Section 2.1, paragraph 3, from line 151.
Please, provide selection criteria for EMT-associated genes in TCGA database retrieving (1.4. EMT and epigenetics…, page 3). It is good idea to present EMT genes/proteins classification, e.g. markers, transcription factors, signaling pathway-related. I suggest to give the full name of all described EMT players and EMT targets; e.g. in figure and table legend.
R: We thank the Reviewer for pointing this out. The entry list of EMT-related players that was introduced in TCGA database for data retrieving was chosen in a way to include both epithelial-prone players (like cadherins) and mesenchymal-prone players (like vimentin) that are well-known to be relevant in the EMT process in cancer and that we frequently study in our Group. The players included are the ones most mentioned and found to be deregulated in the literature review performed by us.
As requested, the full names of the EMT players and the main coding genes were added to the Abbreviations List in page 14.
Please, rethink the section title: (1.4. EMT and epigenetics in bladder cancer, page 3). Do you really discuss EMT and epigenetics in bladder cancer?
R: We fully agree with the reviewer and changed the section title to “Influence of EMT major players in Bladder Cancer”.
I suggest to remove the selection criteria of EMT and epigenetics related publications (page 3, from 113th row). In my opinion this information seems to be redundant. Unless you have performed critical review. In this way, you should present number of articles you have taken for analysis and articles you have removed.
R: As requested the full paragraph was removed.
Table 1. Please, give explanation of: 5AZA, TSA, BSP, FFPE, FF.
R: The explanations were added to Table’s legend.
Table 1 and Table 2. In author or Ref. column you should provide also appropriate number according to references list.
R: The information was added as requested.
Reviewer 2 Report
This is a well structured and presented review about epigenetic mechanisms and their influence to EMT in Bladder Cancer. The references selected also adequately reflect current knowledge about this topic. I believe this work could be accepted for publication after a linguistic polishing.
Exemples:
Line 52: organism - organisms'
Line 78: answer - to answer
Line 82: their their - their
Line 83 processed - process
Line 97: metastization - metastasis or metastatic potential
Line 106: weresignificantly - were significantly
Author Response
This is a well structured and presented review about epigenetic mechanisms and their influence to EMT in Bladder Cancer. The references selected also adequately reflect current knowledge about this topic. I believe this work could be accepted for publication after a linguistic polishing.
Exemples:
Line 52: organism - organisms'
Line 78: answer - to answer
Line 82: their their - their
Line 83 processed - process
Line 97: metastization - metastasis or metastatic potential
Line 106: weresignificantly - were significantly
R: The English language was fully revised throughout the manuscript. All the corrections were performed, as requested.